# Isolated Avulsion Fracture of the Tibial Tuberosity in an Adult Treated with Suture-Bridge Fixation: A Rare Case and Literature Review

**DOI:** 10.3390/medicina59091565

**Published:** 2023-08-29

**Authors:** Dong Hwan Lee, Hwa Sung Lee, Chae-Gwan Kong, Se-Won Lee

**Affiliations:** 1Department of Orthopedic Surgery, Yeouido St. Mary’s Hospital, College of Medicine, The Catholic University of Korea, 10, 63-Ro, Seoul 07345, Republic of Korea; ldh850606@naver.com (D.H.L.);; 2Department of Orthopaedic Surgery, Uijeongbu St. Mary’s Hospital, College of Medicine, The Catholic University of Korea, 271, Cheonbo-Ro, Uijeongbu-si 11765, Republic of Korea

**Keywords:** tibial tuberosity, avulsion fracture, suture bridge fixation, surgical technique

## Abstract

*Background and objectives:* Isolated tibial tuberosity avulsion fractures are exceptionally uncommon among adults, with limited instances documented in published literature. Here, we describe a case of an isolated tibial tuberosity avulsion fracture in an adult that was treated successfully with the suture bridge repair technique. *Patient concerns:* A 65-year-old female visited the outpatient department with left knee pain after a slip and fall. Lateral radiographs and sagittal MR images of the left knee revealed the tibial tuberosity avulsion fracture, but the fracture line did not extend into the knee joint space. Surgical intervention was performed on the patient’s knee using an anterior midline approach, involving open reduction and internal fixation. The avulsed tendon was grasped and pulled, and an appropriate suture location was identified. Using a suture hook, the suture was guided through the patellar tendon as near to its uppermost point of the fragment as achievable, and tied over tendon. A single suture limb from each anchor was fastened over the tibial tuberosity to the distally positioned foot print anchor, effectively anchoring the tibial tuberosity using the suture bridge technique. The patient started walking on crutches after one week and was able to walk independently with a brace after two weeks from the operation day. After three months, the patient had regained her mobility to the level prior to the injury and exhibited painless active range of motion from 0 to 130 degrees. Hardware positioning and bony union were maintained at the one-year follow-up. *Conclusions:* In our case, the open suture bridge fixation method for tibial tuberosity avulsion fractures produced satisfactory results. Open suture bridge fixation may be considered for isolated tibial tuberosity avulsion fractures in adults, especially when the avulsion tip is too small for screw fixation.

## 1. Introduction

Though they comprise less than 3% of all physical injuries in children, tibial tuberosity avulsion fractures are exceptionally uncommon among adults, with limited instances documented in published literature [1,2,3,4,5,6,7]. Tibial tuberosity fractures are mainly treated by two methods: using lag screws to secure the main fragment from anterior to posterior direction and tension band wiring with cerclage wires at the patella tendon insertion site on the tibial tuberosity [8]. The suture bridge fixation technique was first introduced as an effective method for repairing rotator cuff tears. By promoting a broad contact surface, healing is improved and results in stronger fixation. Its application has subsequently expanded to the greater tuberosity avulsion fractures of the humerus [9]. Moreover, the employment of the suture bridge repair technique for surgical intervention in Achilles insertional tendinopathy is widely recognized [10,11], and a suture bridge fixation approach has also been introduced for Achilles tendon avulsion fractures of the calcaneus [12]. Additionally, a similar approach has been employed in cases involving the triceps insertion at the elbow [13]. Thus, the suture bridge repair technique for avulsions at large tendon insertion sites across the body appears to be considered an effective therapeutic approach.

However, to the best of our knowledge, this technique has yet to be adopted for treatment of tibial tuberosity avulsion fractures in adults. Here, we describe a case report in which the suture bridge technique was adopted to repair an isolated tibial tuberosity avulsion fracture in an adult. Additionally, a literature review of previously reported cases of tibial tuberosity avulsion fracture was conducted to emphasize the advantages of fixation using the suture bridge technique. Furthermore, the rehabilitation protocols outlined in the existing reports were reviewed and compared to our approach.

## 2. Case Presentation

A 65-year-old female visited the outpatient department with left knee pain after a slip and fall. The patient had a past history of surgery with tension band wiring for a left (ipsilateral side) displaced patellar fracture one year prior. The hardware had been removed six months before the present injury. The patient’s plain radiography of the left knee exhibited osteoporotic bone quality in the patella, distal femur, and proximal tibia, which is presumed to be a result of previous trauma and two prior surgeries. She had difficulty in performing active knee extension and complete leg straightening. Lateral radiographs and sagittal MR images of the left knee revealed the tibial tuberosity avulsion fracture, but the fracture line did not extend into the knee joint space (Figure 1).

Surgical intervention was performed on the patient’s knee using an anterior midline approach, involving open reduction and internal fixation. The avulsed tendon was grasped and pulled, and an appropriate suture location was identified. In the knee-full-extension state, No. 2 Ethibond sutures were applied for temporary traction while two metal suture anchors (5.5 mm TwinFix; Smith & Nephew, Andover, MA, USA) were inserted 15 mm from each other. A temporary K-wire was inserted under the image intensifier for fixation of the fragment in its anatomical position. Using a suture hook, the suture was guided through the patellar tendon as near to its uppermost point of the fragment as achievable and tied over tendon. After that, a single suture limb from each anchor was fastened over the tibial tuberosity to the distally positioned 5.5-mm Footprint anchors (Smith and Nephew, Andover, MA, USA), effectively anchoring the tibial tuberosity using the suture bridge technique (Figure 2 and Figure 3). To achieve firm fixation, maintaining proper tension during the crossing of suture limbs and fixation is crucial, along with the placement of the anchors. The temporary K-wire and traction sutures were subsequently removed. Postoperatively, the patient was placed in a functional brace that limited up to 90 degree flexion for four weeks.

The patient initiated passive range of motion (ROM) exercises one day after surgery. The range of motion was gradually increased, with a restriction of up to 90 degrees for the first 4 weeks. The patient was permitted to bear weight according to their comfort level. She started walking on crutches after one week from the operation day and was able to walk independently with a brace after two weeks. Until the sixth week, only ambulation on level ground was allowed, and, after 6 weeks, the patient started ambulation with active stretching, including stair climbing. After three months, the patient had regained her mobility to the level prior to the injury and exhibited painless active range of motion from 0 to 130 degrees (Figure 4). On the 12-month postoperative X-ray, the position of the metal anchor remained unchanged, and the fracture fragment was well-maintained without any displacement and had successfully achieved union (Figure 5).

## 3. Discussion

Isolated tibial tuberosity avulsion fractures are exceptionally uncommon among adults. Only six previous case were documented in the published literature [1,2,3,5,6,7]. In adolescents, they are a kind of growth plate injury and make up less than 3% of all such injuries [14,15,16,17]. They are believed to occur due to sudden quadriceps contraction against knees that are partially flexed, like during the landing after a fall or jump [4,18,19].

The closing growth plate during adolescence is vulnerable to pulling forces and can be easily avulsed [20]. In adults, the exact cause is not clear. In general cases, sudden contraction of the quadriceps muscle may result in injury to weaker structures such as the quadriceps or patellar tendons. However, in patients with osteoporotic bone quality, it appears that the bone in the tibial tuberosity becomes weak, leading to fractures in that area [3]. Moreover, even in cases of good bone quality, tibial tuberosity fractures can occur with high-energy traumas, such as falling from a height. Another possibility is to consider that avulsion fractures can occur due to direct trauma towards the tibial tuberosity. In the previous six documented cases in adults, one fell from a ladder, and two suffered direct injuries to the tibial tuberosity. The other three cases involved relatively low-energy traumas, specifically fall and twisting injuries. Among them, two cases were observed in super-elderly patients, while the remaining case was associated with Paget’s disease, indicating that all cases occurred in osteoporotic bone (Table 1). In our case, avulsion of the tibial tuberosity is theorized to have occurred due to forceful full flexion of the knee, prompted by patellar baja caused by the previously healed patellar fracture. We believe that this is a rare complication that can occur at the end stage of full flexion in patellar fractures in adults and was not previously reported.

Various techniques for surgically treating avulsion fractures of the tibial tuberosity have been reported for individuals across all age groups. The AO Foundation suggests two techniques for fixation of tibial tuberosity fractures [8]. One technique uses lag screws to secure the main fragment from anterior to posterior direction [20]. The other method utilizes tension band wiring with cerclage wires inserted through the Sharpey’s fibers at the patellar tendon insertion site on the tibial tuberosity [3]. In our case, the main fragment was too small for either screw fixation or tension band wiring. Therefore, we considered the use of suture anchors for fixation and decided to perform the suture bridge technique, which we deemed more suitable for fixation of an avulsed fragment. 

In the majority of previously reported cases, patients were advised to restrict weight-bearing for an extended period after the surgery. Furthermore, in the majority of cases, the initiation of knee range-of-motion exercises was delayed, and the duration of immobilization was extended. Among the six reported cases, in four of them, immobilization was maintained using a cast or brace for approximately 4–6 weeks. In older patients, like ours, this could lead to decreased physical fitness and atrophy of the quadriceps muscles. Furthermore, these older patients often have a weaker motivation for rehabilitation and may find it challenging to endure a painful rehabilitation process. Therefore, if immobilization is prolonged, there is a higher risk of developing knee stiffness. The potential advantages of using suture bridge fixation are early rehabilitation and weight-bearing. It took one week for our patient to start walking on crutches and two weeks to walk by herself with a brace applied on the affected knee. She regained her mobility to the level prior to the injury and exhibited painless active range of motion from 0 to 130 degrees at 12 months after surgery. Among the previously reported cases, Choi et al. utilized suture anchors and employed a different fixation approach from ours for treatment [2]. 

They reported applying a functional brace immediately after surgery, allowing 0–30° of motion in the first week, and gradually increasing the range of motion through rehabilitation. By the fourth week, they were able to achieve a knee motion of 0–120°. Considering our case alongside theirs, the use of suture anchors for fixation appears to provide sufficient fixation power to enable early ambulation and range-of-motion exercises without problems. However, even when utilizing alternative methods, such as screws, plates, or tension band wiring, if a sufficiently secure fixation can be achieved, early ambulation and range-of-motion exercises can be performed without any difficulties or issues. Regarding ambulation, we believe that weight bearing itself does not pose a problem for the fracture recovery, as long as early activities involving active stretching, such as climbing stairs or steep inclines, are avoided. Therefore, in cases where the fracture fragment is large, using plates, screws, or tension band wiring for surgery, it is recommended to ensure firm fixation during the surgery and initiate early rehabilitation after the procedure.

Another advantage of using the suture bridge technique for repair is its usefulness in stabilizing small bone fragments. In addition, the advantage of our proposed procedure is that there is no need for additional surgery to remove any inserted instrument. Although the removal of instruments is not a major surgery, it can provide a great advantage in terms of relieving patients from the burden of surgery. Given these advantages, the suture bridge technique could become a valuable option for surgical management of tibial tuberosity avulsion fractures in adults, especially when the avulsion tip is too small for screw fixation.

## 4. Conclusions

In our case, the open suture bridge fixation method for tibial tuberosity avulsion fractures produced satisfactory results. Open suture bridge fixation may be considered for isolated tibial tuberosity avulsion fractures in adults, especially when the avulsion tip is too small for screw fixation. After surgery, early ambulation and early range-of-motion exercises are possible and recommended. Through early rehabilitation exercise, optimal therapeutic outcomes can be achieved.

## Figures and Tables

**Figure 1 medicina-59-01565-f001:**
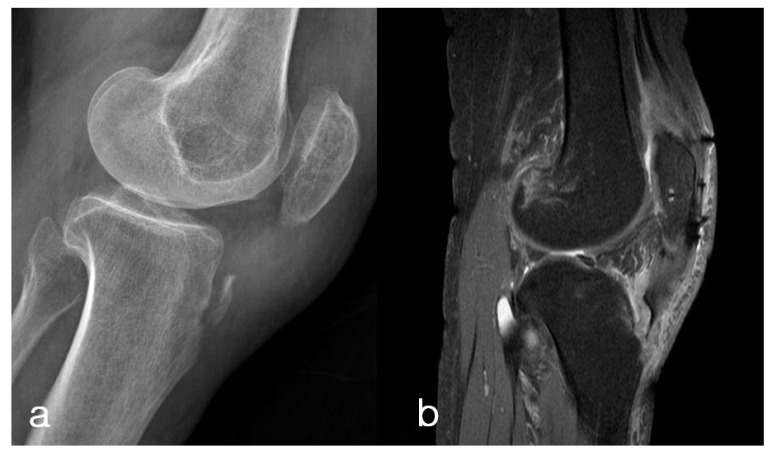
Lateral radiograph (**a**) and sagittal MR image (**b**) of the left knee, revealing the tibial tuberosity avulsion fracture, but the fracture line does not extend into the knee joint space. Plain radiography exhibited osteoporotic bone quality in the patella, distal femur, and proximal tibia, which is presumed to be a result of previous trauma and two prior surgeries.

**Figure 2 medicina-59-01565-f002:**
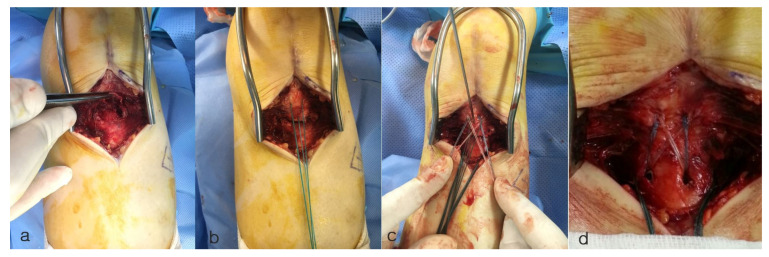
Intra-operative images. (**a**) Image depicts a tibial tuberosity avulsion fracture, as indicated by the forceps holding the avulsed fragment. (**b**) The avulsed tendon was grasped and pulled using No. 2 Ethibond suture, and an appropriate suture anchor location was identified. (**c**) Image shows the fixation method using the suture bridge technique. A single suture limb from each anchor was secured over the tibial tuberosity and fastened to the distally positioned footprint anchors. (**d**) A well-reduced state was observed with the suture bridge repair technique.

**Figure 3 medicina-59-01565-f003:**
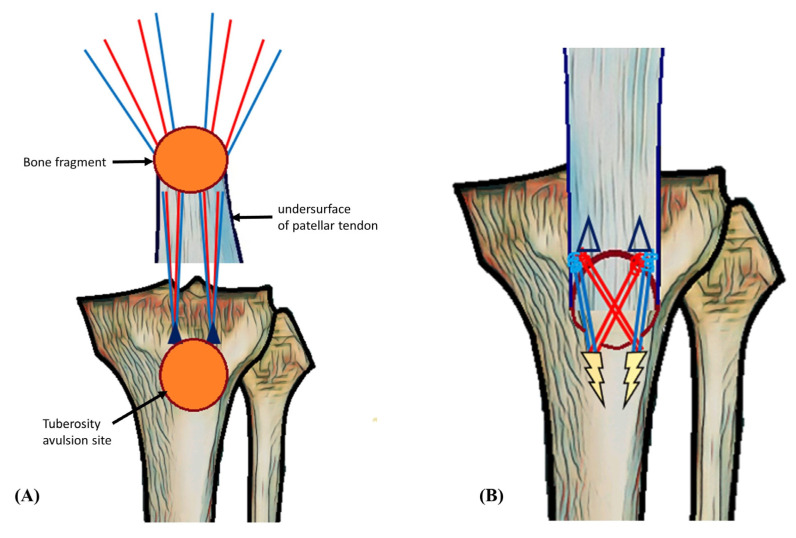
Schematic image of surgical techniques. (**A**) An image viewed from the undersurface of the patellar tendon showing the passage of the suture. Two orange circles indicate the fracture surfaces. Two navy blue triangles indicate the position of the 5.5 mm metal anchors. The red and blue lines indicate the sutures that are attached to the anchors. (**B**) An image depicting the appearance after suture bridge fixation. Two yellow markers indicate the positions of the 5.5 mm footprint anchors. The image shows the crossing of suture limbs from each anchor to secure the avulsed tibial tuberosity fragment using the distally positioned footprint anchor. To achieve firm fixation, maintaining proper tension during the crossing of suture limbs and fixation is crucial, along with the placement of the anchors.

**Figure 4 medicina-59-01565-f004:**
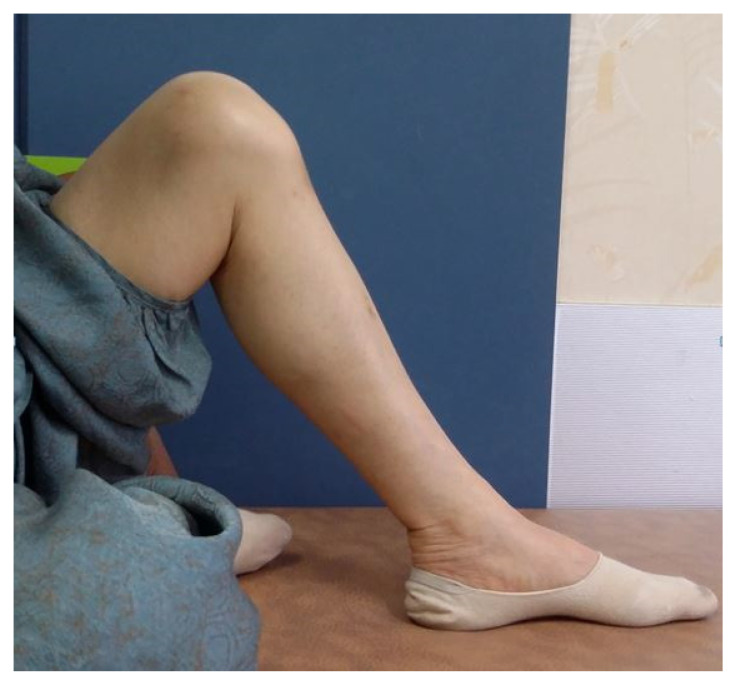
Clinical image of the patient’s lower extremity with recovery of range of motion at three months after the operation.

**Figure 5 medicina-59-01565-f005:**
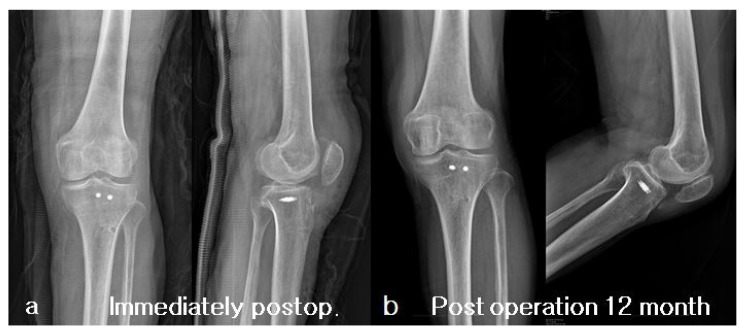
Post-operative X-rays. (**a**) Immediate postoperative X-rays show that the avulsed tibial tuberosity fragment has been successfully reduced to its anatomical position. (**b**) 12-month postoperative X-rays show that the fracture fragment is well-maintained without any displacement and has successfully achieved union.

**Table 1 medicina-59-01565-t001:** Review of reported isolated tibial tuberosity avulsion fractures.

	Age (Years)	Injury Mechanism	Fixation Method	Remarks
Mounasamy et al. [5]	49	Fall from ladder	Plate and screws	Immobilized in a knee brace for three weeks
Pires et al [6].	62	Direct injury	Lag screw	Immobilized with a long brace for six weeks
Choi et al. [2]	67	Direct injury	Two suture anchor and fiber-tape	Functional brace for four weeks; 0 to 30° at first week, 0 to 120° at four weeks
Brown et al. [1]	86	Fall with twist	Screw and spike	Second fall caused implant loosening.
			Tendon repair using suture anchor at the second surgery.	Immobilized with a cylindrical cast for four weeks and changed to a genu brace.
K AJ et al. [3]	88	Fall with twist	Lag screw and tension band wiring	Immobilized with an extension splint for six weeks and tolerable weight bearing.
Raad et al. [7]	54	Fall with twist	Lag screw and tension band wiring	Underlying Paget’s disease. Immobilized with a cast for six weeks and non-weight bearing. Changed to functional brace with progressive weight-bearing until 12 weeks.

## Data Availability

All data concerning the case are presented in the manuscript.

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
