# Peer review of "Isolated Avulsion Fracture of the Tibial Tuberosity in an Adult Treated with Suture-Bridge Fixation: A Rare Case and Literature Review"

_medicina, 2023, doi:10.3390/medicina59091565_

Round 1
Reviewer 1 Report
Great case - in most parts good presentation and especially good review of formerly presented or published cases. Even your pictures are great nethertheless a small drawing would make it more easy to repeat your procedure for the readers in similar cases. You should maybe also clearify the postop protocol a little bit more - did you really allow aktive stretching the knee - like walking on stairs - with a range of 0/0/90 after 2 weeks ?
Reviewer 2 Report
As it has been some time since I performed orthopedic surgery, I ask a Harvard-trained colleague to review the technique described in this article. He had one identical case in his 20 years in practice and performed the same procedure as described in the article with fast mobilization of the patient. This reinforces the author's conclusions in regard to the open suture bridge fixation method as an excellent approach to tibial tuberosity fractures.
- This is a case study of an unusual problem seen in the orthopedic world
- The surgical approach in treating this condition is novel and appears to work, not only by the presenters, but also a colleague of mine in the business 20 years who had a similar case and treated it the same way,
- It shows the value of using sutures to stabilize the fracture with relative rapid ambulation, especially important in elderly people.
- This is a single case study of a relatively rare fracture. No further commentary is needed.
- The article's use of sutures for this fracture is appropriate and is a method that should be considered when confronting the problem.
- The references seem appropriate
- No further comments on my part
Author Response
Thank you for providing such a valuable comment.